# Comparison of External and Internal Training Loads in Elite Junior Male Tennis Players During Offensive vs. Defensive Strategy Conditions: A Pilot Study

**DOI:** 10.3390/sports13040101

**Published:** 2025-03-26

**Authors:** Péter János Tóth, Gabriella Trzaskoma-Bicsérdy, Łukasz Trzaskoma, János Négyesi, Károly Dobos, Krisztián Havanecz, Sándor Sáfár, Csaba Ökrös

**Affiliations:** 1Department of Sport Games, Hungarian University of Sports Science, 1123 Budapest, Hungary; karoly.dobos93@gmail.com (K.D.); okros.csaba@tf.hu (C.Ö.); 2Institute of Sport Science, Eszterházy Károly Catholic University, 3300 Eger, Hungary; bicserdy.gabriella@uni-eszterhazy.hu; 3Department of Theory of Sport, Józef Piłsudski University of Physical Education in Warsaw, 00-968 Warsaw, Poland; lukasztrzaskoma77@gmail.com; 4Department of Kinesiology, Hungarian University of Sports Science, 1123 Budapest, Hungary; negyesi.janos@tf.hu; 5Training Theory and Methodology Research Center, Hungarian University of Sports Science, 1123 Budapest, Hungary; havanecz.krisztian@tf.hu (K.H.); safar.sandor@tf.hu (S.S.)

**Keywords:** racket sports, playing style, GPS, RPE, acute/chronic workload ratio

## Abstract

The aim of our pilot study was to investigate the effects of offensive and defensive strategy conditions on external and internal training load factors in male tennis players. This study included six elite junior male tennis players (chronological age: 15.7 ± 1.0; body height: 180.7 ± 6.5 cm; body mass: 71.0 ± 10.8 kg) who had to play two simulated matches. Among the external training load variables, running activities were measured with a GPS sensor operating at 10 Hz and a 100 Hz tri-axial piezoelectric linear accelerometer integrated into it; furthermore, tennis shot activities were measured with a tennis racket-mounted smart sensor. Internal training load was measured subjectively using the RPE method. The results show that players scored significantly higher on the PlayerLoad (*p* = 0.031; r = 0.90) and IMA CoD low right (*p* = 0.031; r = 0.90) running variables and on the forehand spin (*p* = 0.031; r = 0.90) and backhand spin (*p* = 0.031; r = 0.90) when using a defensive strategy. There were no significant differences between the two strategy conditions in all other external and internal training load parameters. The defensive strategy has more acceleration in all three planes of motion, suggesting that conditioning training should be placed in the intermittent endurance capacities for players who predominantly use this strategy.

## 1. Introduction

The outcome of tennis matches is often extremely unpredictable. This unpredictability is also caused by the different strategies, technical–tactical decisions, state of readiness, playing surfaces, and weather conditions of the players [1]. This naturally leads to variations in match times, which can last less than an hour but can also be seen in some cases lasting more than five hours [2]. According to the International Tennis Federation (ITF) rules, high-intensity periods are separated by rest periods of predetermined duration [3], according to which most high-level matches have a work-to-rest ratio of between 1:2 and 1:5, with point durations ranging from 3 s to 15 s on average [4,5]. In addition, matches are characterized by high-intensity micromovements (accelerations, decelerations, change of directions, and jumps) as running activities. Analyzing these movements, it can be seen that players execute 80% of strokes after positional movements within 2.5 m, 10% of strokes are performed between 2.5 and 4.5 m, while 5% of strokes are performed over 4.5 m, and the remaining 5% do not reach the ball [6,7]. The average distance run within each rally is 6–7 m [8], with the maximum distance run between strokes ranging from 8 to 12 m [9] and an average of 4 to 6 changes of directions per point, which applies to a single player [3]. This type of approach to match analysis is called a reductionist method in ball games and means that physical actions are analyzed in isolation, separately from tactics and techniques [10]. Individual physical actions are modulated by tactical movements during the game, where players adopt several options defensively or offensively according to different tactical contexts [11]. In tennis, this integrated approach is based, among others, on the various playing styles that are associated with a dominant offensive or defensive strategy conception [12]. In itself, effective playing time differs between styles, with 21% for offensive players, 28.6% for all-court players, and 38.5% for baseline players when tested on clay surfaces compared to total playing time [13]. Although playing styles and strategies in tennis are seldom clearly defined or studied [12], two opposing approaches can be observed in practice: At the highest levels of play, the number of winners is almost equal to the number of errors. Therefore, a common strategy (defensive strategy) for success is to minimize errors by playing passively from the baseline. The opposite strategy (offensive strategy) involves dominating rallies through active play, using powerful topspin strokes at sharp angles across the entire court. This strategy aims to force the opponent to make errors or win points directly [14].

In female tennis players, the evolution of technical–tactical actions, external and internal training load factors, and activity profiles in closed conditions for each strategy condition has been investigated [12]. The results showed that the largest effects compared to the own (control) condition were found in technique and tactics (number of groundstrokes and errors) and activity profile (strokes per rally, rally duration, work-to-rest ratio, and effective playing time), followed by external load parameters (high- and low-intensity running distance). From this it can be seen that techniques–tactics will be affected first by the strategy decision, and only then will the external training load change. In addition to this, another important aspect is the study of the internal training load, which represents the athlete’s internal response to the external load. In tennis, this is measured by the heart rate (HR) variables [1,15], VO_2_ [3,4], blood lactate [16], and rating of perceived exertion (RPE) [17,18]; data are commonly used to characterize it. The subjective RPE-based method is easy to use, integrating physiological, mechanical, and neuromuscular fatigue [19] and showing a strong correlation (r = 0.74) with HR [20]. In tennis matches, the average RPE score ranges from 10 to 16 (Borg’s CR-20 scale) [3,21] and 5–8 (Borg’s CR-10 scale) [22], but this may be influenced by external factors such as playing surface [23], skill level [24], gender [25], and game situation [18,21]. Among the game situations, the serve and return situation is commonly analyzed, showing that serve games are more demanding than return games [18,21]. In addition, different training conditions are also analyzed in terms of external and internal load. For instance, coach-fed drills are the most demanding in terms of external training load indicators such as the number of strokes and average racket velocity. Meanwhile, racket-fed drills have the greatest impact on the total distance covered by players and their average speed [26].

As with the external load parameters, it is also useful to consider RPE indicators in an integrated way with different strategies and tactics. Previous research in female tennis players has indicated that RPE was the highest when using the passive strategy condition [27]. In open conditions, where players play according to their own style, table tennis players were studied for differences in activity profile and internal load between attacking and all-court players in simulated matches [28]. The results showed no significant difference in energy production between the two playing styles, but attacking players performed significantly more strokes (*p* = 0.03). Besides all this, these training load monitoring procedures can also be used in practice to reduce the risk of sports injuries, for which the researchers developed the Acute/Chronic Workload Ratio (ACWR) method [29,30]. The size of the current weekly training load (called acute training load) in relation to the longer-term training load (called chronic training load) determines the ACWR, and when the ACWR is within the range of 0.8–1.3, the risk of injury is relatively low. However, if the ACWR was ≥1.5 (i.e., the acute training load was much greater than the chronic training load), the risk of injury significantly increases [31].

To the best of our current knowledge, no research has yet been conducted that has examined the effects of different strategy conditions on external and internal training load variables in male tennis players, and, therefore, we sought to address these in our current pilot study. The main objective of our research was to compare the differences between offensive and defensive playing strategies under controlled conditions in terms of the most commonly used external and internal training load indicators in tennis. Based on prior literature and coaching experience, we hypothesized that there would be a difference between simulated matches played in offensive and defensive strategy conditions in (i) external training load variables in running and tennis shot activities, but (ii) no significant difference in internal load factors like subjective RPE in elite junior male tennis players.

## 2. Materials and Methods

### 2.1. Experimental Approach

In this cross-sectional, applied, pilot study, we identified differences between the two strategy conditions in the most relevant external and internal load parameters in tennis. This study was a one-day study, conducted in May 2023, and took place between 2:00 p.m. and 4:00 p.m. All players were from a Hungarian tennis club; thus, we could perform the measurements under their own conditions. The experiment took place on an outdoor, windless, clay court (temperature: 21.2–24.5 °C; relative humidity: 44–56%; Kestrel 4000 Pocket Weather Tracker, Nielsen Kellerman, Boothwyn, PA, USA).

### 2.2. Participants

In this pilot study, we included 6 elite junior male tennis players (chronological age: 15.7 ± 1.0; body height: 180.7 ± 6.5 cm; body mass: 71.0 ± 10.8 kg), selected by theoretical sampling. Inclusion criteria are as follows: (i) U16-U18 male age group, (ii) national age group ranking ≤10, (iii) right-hand dominant, and (iv) ability to apply both offensive and defensive playing style. The exclusion criteria are as follows: (i) injury/illness, (ii) having undergone some type of orthopedic surgery in the last 12 months, and (iii) having performed high-intensity activity in the last 48 h. Notably, after adhering to the exclusion criteria, there are a relatively small number of participants included in this study. Nevertheless, our study is strengthened by the fact that the final participants were elite junior or already professional-level tennis players, ensuring that very high-quality matches took place.

The players in this study participated in an average of 10.3 ± 1.5 h of tennis trainings and 3.3 ± 0.7 h of conditioning trainings per week in a microcycle. In addition, they averaged 23.1 ± 4.5 official national and/or international competitions per season. Of the participants, one player was ranked in the top 200, one player in the top 500, and the other players were ranked in the top 2000 in the official junior rankings of the International Tennis Federation (ITF) at the time of the experiment, and two of these players were already ranked in the Association of Tennis Professionals (ATP) singles ranking. The participants were all all-court players, meaning that they were confident in using both offensive and defensive strategies [13].

Prior to the start of the experiment, the players and their parents/legal guardians were informed of the research process, and their written consent was obtained. The local institutional ethics committee (Hungarian University of Sports Science, Budapest, Hungary; Approval No. TE-KEB/02/2022; Approval date: 6 February 2022) approved the procedures in accordance with the latest version of the Declaration of Helsinki.

### 2.3. Procedures

Figure 1 shows the schematic illustration of the experimental design. The participants were aware of the procedure, as they had attended a pre-briefing. The experiment day started with general anthropometric assessments, followed by simulated matches. Body height was measured using a fixed stadiometer (±0.1 cm; Holtain Ltd., Crosswell, UK) and the body mass with a digital balance (±0.1 kg; ADE Electronic Column Scales, Hamburg, Germany). After the anthropometric assessments, the players performed a general warm-up (15 min) consisting of low-intensity circulatory exercises, muscle activation, dynamic stretching, and neuromuscular activation exercises. They then proceeded to the tennis court with a tennis-specific warm-up protocol (10 min), in which they performed groundstrokes, volleys, serves, and returns. The tennis players were then divided into pairs based on the level of play, which had been previously conducted by their own coaches. Participants were asked to play two 10-min simulated matches with 5 min of passive rest according to a pre-described protocol [12,27], which was developed by Hoppe and colleagues [12,27] based on preliminary tennis-specific match analysis studies [3,18]. The 10-min intervals were chosen because they allowed the players to play multiple matches in one day without the influence of accumulating fatigue [12,27]; furthermore, the players were between two tournaments and could not be overloaded, as they are professional athletes and they were in season.

The matches were played according to the ITF official rules in a tie-break format, meaning that after the first serve, the serve was exchanged in deuce. For the simulated matches, brand-new 53–56 g and 6.5 diameter ‘Slazenger Ultra Vis’ balls were used that met international standards. During the matches, tennis players had to pick up the balls and count the points themselves. Before each of the two 10-min blocks, the players were instructed on whether to play the match with an offensive or defensive strategy. Each instruction was given on paper and in an open format. For the offensive strategy, the players read ‘try to win the point by yourself or by forcing the opponent to make a mistake’, and for the defensive situation, the participants read ‘try to win the point by reducing the number of unforced errors’. With these brief yet concise instructions, the players applied tactical situation characteristics of the respective strategies, as mentioned earlier [14]. These instructions were given in Hungarian, as the players were of Hungarian nationality. Furthermore, the instructions were clear to the players, as they had encountered them multiple times during their regular tactical training sessions. The players were not allowed to talk to each other about who had received what instructions. Each player was required to play a match with both an offensive and defensive strategy, and the pairings were always set up so that two players in offensive and two players in defensive roles played against each other. Overall, we had 6 simulated matches with 6 offensive and 6 defensive strategy conditions. Halfway through the matches, the tennis players were always warned by a coach to follow the instructions given, and if the player failed to follow the given instructions, the ITF-qualified coach, who closely monitored the match, immediately warned the player. After 10 min, the last point was played.

### 2.4. Variables

#### 2.4.1. External Training Load

For the external training load, both running and tennis shot activities were investigated. To assess running activities, we used a GPS sensor operating at 10 Hz (OptimEye S5; Catapult Innovations, Melbourne, Australia) and a 100 Hz tri-axial piezoelectric linear accelerometer (Kionix: KXP94) integrated into it. In addition to the accelerometer data, the gyroscope and magnetometer integrated into the microsensor also help to determine the direction of each acceleration [27]. This type of GPS sensor has good reliability and accuracy in the study of movements over small areas [32,33] and also has good reliability in the study of tennis-specific movements [34]. The sensors were worn by the players between their two shoulder blades in a neoprene vest. The players were familiar with the use of the sensors, as they had previously used such a measurement device in their everyday training. The harness size was chosen by players in training sessions. The sensors were switched on 15 min before the measurements started but were only inserted into the vests at the start of the simulated matches. In the present study, we were directly interested only in the data collected by the inertial measurement unit (IMU) embedded in the GPS, which investigates high-intensity micromovements, since these are the most frequent running activities in tennis [35]. For the parameters under investigation, we only considered absolute values. To measure the individual micromovements, we used the PlayerLoad (PL) variable, which sums the acceleration in all directions of the court by a number, taking into account the instantaneous rate of change in acceleration and dividing it by the scale factor (divided by 100) [36] to obtain an arbitrary unit (AU) number. Reliability of the PL metric has previously been established at a 1.9% coefficient of variation (CV) from observations in team sport athletes [37]. In addition to this, we also investigated the number (n) of right and left change of directions (IMA CoD low and high) with low (<2.5 m/s^2^) and high (≥2.5 m/s^2^) intensity. Rightward changes of direction are defined as accelerations between 45° and 135°, and leftward changes of direction as accelerations between −135° and −45° [34]. These inertial movement analysis (IMA)-type data have been previously used in tennis-specific research to determine the direction of accelerations [34]. All data recorded by the Catapult units were downloaded and elaborated by Catapult software (OpenField v1.22.2; Catapult Innovations, Melbourne, Australia) before being exported as a .CSV file for further analysis.

To measure tennis shot activities, we used a tennis racket-mounted smart sensor (Zepp Tennis 2.2.1, Zepp Labs, San Jose, CA, USA) with a dual accelerometer and a dual 3-axis gyroscope. The validation procedure of this sensor showed good results for the number of strokes and ball speed and moderate results for the other types of strokes [38]. The shot velocity measurements from the Zepp sensor showed an almost perfect agreement with the gold standard VICON system (ICC = 0.983; *p* < 0.001), demonstrating excellent validity for speed measurement [38]. In addition, this type of smart sensor showed moderate agreement with actual shot types (κ = 0.612), like the spin parameters [38]. Currently, it is the only smart sensor that is compatible with all rackets [26,39]. The smart sensors are mounted on the end of players’ rackets using a manufacturer-provided ‘flex-mount’ adapter that can be easily moved from one type of racket to another and thus reduce interference caused by different grips [40]. Players were also aware of the use of smart sensors, and player profiles with sociodemographic data were created beforehand, through which the sensors and the associated phone app (Zepp Tennis, Zepp Inc., Milpitas, CA, USA) were linked. Each player used the same sensor for each of the two simulated matches in order to avoid data dissociation, and each sensor was connected to a separate smartphone on a Bluetooth basis to avoid interference. Among the tennis shot variables, we examined (i) forehand velocity (km·h^−1^), (ii) backhand velocity (km·h^−1^), (iii) forehand spin (revolutions per minute, rpm), and (iv) backhand spin (revolutions per minute, rpm). All data were detected in the phone application and then exported to .CSV file format for further analysis.

#### 2.4.2. Internal Training Load

To examine the internal load, we measured the players’ rating of perceived exertion (RPE) (Borg’s CR-10 scale). The validity of this RPE scale shows a strong correlation with heart rate (r = 0.74; *p* < 0.001) and blood lactate (r = 0.83; *p* < 0.001) during aerobic exercise [41]. Even in small field sports such as tennis, RPE measurements are often used to assess the internal exertion of athletes [15,17]. Immediately after each simulated match, RPE data were recorded following a preliminary protocol [42] (Figure 1). Players were asked, ‘How demanding was the match?’ and were asked to score this on a 0–10 Likert scale. The obtained RPE results were manually recorded on paper, which were then entered into spreadsheet software (Microsoft Excel, 16.49, Microsoft Inc., Washington, DA, USA) on a computer and finally used in CSV. file format for further analysis.

### 2.5. Statistical Analyses

Statistical analyses were performed using SPSS Statistics Package (version 20.0, SPSS Inc., Chicago, IL, USA). First of all, all data’s distributions were checked by Shapiro–Wilk’s test, kurtosis and skewness values, and visual inspection of their histograms and QQ plots. Descriptive statistics are reported as mean ± standard deviation (SD). Since none of the dependent variables followed a normal distribution, a Wilcoxon signed-rank test was used to determine the differences between the two strategy conditions (offensive and defensive). Effect sizes (ES) were interpreted using the r principle as follows: very small <0.1, small 0.1–0.3, medium 0.3–0.5, and large >0.5 [43]. The significance level was set at *p* < 0.05.

## 3. Results

Table 1 shows the descriptive statistics of the external and internal load variable data. In terms of running activities, there is a significant difference with a large effect size between the offensive and defensive strategy conditions for the PL (T = 0.0; Z = 2.201; *p* = 0.031; r = 0.90) (Figure 2) and IMA CoD low right (T = 0.0; Z = 2.201; *p* = 0.031; r = 0.90) (Figure 3) parameters. No significant difference was observed between the two strategy conditions for the IMA CoD low left (T = 2.0; Z = 1.782; *p* = 0.090; r = 0.73), IMA CoD high right (T = 3.0; Z = 1.572; *p* = 0.140; r = 0.64), and IMA CoD high left (T = 12.0; Z = 1.214; *p* = 0.279; r = 0.50) parameters. For the tennis shot activities, a significant difference was observed with a high effect size between the offensive and defensive strategy condition in the forehand spin (T = 0.0; Z = 2.201; *p* = 0.031; r = 0.90) (Figure 4) and in the backhand spin strokes (T = 0.0; Z = 2.201; *p* = 0.031; r = 0.90) (Figure 5). No significant differences were observed in the forehand velocity (T = 6.0; Z = 0.365; *p* = 0.855; r = 0.15) and in the backhand velocity (T = 16.0; Z = 1.153; *p* = 0.313; r = 0.47). No significant difference was observed between the two conditions in the RPE parameter when testing the internal load (T = 1.5; Z = 1.633; *p* = 0.102; r = 0.67).

## 4. Discussion

The aim of this pilot study was to investigate the difference between the offensive and defensive strategy conditions in external load factors such as running and hitting activities and internal load factors such as the rating of perceived exertion (RPE). We hypothesized that there would be a significant difference between the two conditions in (i) running and tennis shot activities but (ii) no difference in RPE.

In today’s sports science, the analysis of the different load parameters (external and internal) in the analysis of individual ball games is no longer based on a reductionist analysis but on an integrated approach with technique and tactics [10]. In football, for example, this approach can be understood in terms of comparing each position, the most intensive period of the match, and general and specific tactical roles [44]. For example, the distance covered in the high-intensity running zone in a match is the greatest for the wide midfielder position (~800–1500 m) compared to other positions [45,46]. All of this can be further decomposed by including specific tactical roles in the analysis, as this provides the actual physical workload of the athletes [47]. For instance, when a player playing in the central midfielder position is in an attacking role, he is covering a greater distance in the high-intensity running zone than when he is defending [48,49]. This integrated approach is most commonly used in tennis to assess the impact of, for example, matches on different playing surfaces on external and internal load factors [23,24] or various playing situations [18,21] or the profile of different playing styles and strategies [12,13,27].

Success in tennis has its own complexity, which depends on factors such as the athlete’s anthropometric characteristics, physical capacity, technical–tactical skills, psychological characteristics, and medical characteristics [1]. Based on this complexity, many factors influence match data. Research by Murias and colleagues [50] showed that heart rate and lactate levels were higher on clay surfaces than on hard court surfaces, and Reid and colleagues [51] confirmed these, and even RPE was higher on clay surfaces. Another integrated approach to looking at the differences between serve and return games [15] is that they have a higher internal load than return games [18,21], but there can of course be differences depending on the level of knowledge [15]. A third integrated approach in tennis is to look at differences between different playing styles and playing strategies in terms of external and/or internal load factors. In the present pilot study, this was also investigated in a closed setting, as there is little research on this topic and even less among male tennis players. A previous study has investigated the impact of the use of offensive and defensive strategies on match activities and physiological characteristics [4]. The results of this study showed that the defensive strategy resulted in a longer duration of each rally and effective playing time, as well as higher heart rate and blood lactate values (all *p* < 0.01). In the present study, we did not examine the aforementioned load variables, but our results showed that for the defensive strategy condition, significantly higher values were obtained for the PlayerLoad variable, which implies that players in this strategy perform more accelerations in all three planes of motion than in the offensive strategy, and higher values for the IMA CoD low right parameter. This suggests that, since all participants were right-handed in the research, and considering that in modern tennis, more forehand strokes are made than backhands, a statistically significant difference in favor of the defensive strategy occurred in this IMA CoD low right variable. This is because, in this parameter, rallies tend to be longer, and therefore, more shots are played [13,27]. Although there was no statistically significant effect of the different strategies on the other external load variables, large or medium effect sizes were still observed. The running activities in which there was no statistically significant difference can be explained by the fact that, for example, in the two high-intensity direction changes, the players already achieved low values, which are negligible compared to the low-intensity movements. Among these, when examining the IMA CoD right high variable, relatively higher standard deviation values were found despite the small sample size, which we believe may have arisen due to the participants having different anthropometric characteristics (body height and body mass), which can significantly influence the development of high-intensity actions. In our case, no significant difference was observed between the two strategies for an internal load variable such as RPE, which is also supported by previous research [27], but it showed a large effect size in favor of the defensive strategy (*p* = 0.102; r = 0.67). This could mean that, despite the lack of a statistically significant difference between the two game strategies in the RPE variable, it is clear that players became more fatigued in the defensive strategy condition. This is supported by the study of Bernardi and colleagues [13], which found that defensive players have the highest effective playing time, with a 38.5% higher value, which may be correlated with the development of internal load.

It is always the techniques–tactics that will be affected first by the application of different strategies, and only then the tennis shot and running activities, and then, at the very end, what all these will result in, the internal load [52,53]. This statement is supported by the research findings of Hoppe and colleagues [12], who found that the passive, active, and mixed playing strategy conditions induce large effects on external loads (running distances with high acceleration and deceleration), moderate effects on internal loads (energy expenditures with high metabolic power, lactate concentration, and rating of effort), and very large effects on technical–tactical actions (number of ground strokes and errors) and activity profiles (strokes per rally, rally duration, work-to-rest ratio, and effective playing time). In the study by Cayetano and colleagues [26], which examined the effects of different types of tennis drills on external and internal load, the differences between the various drills first manifested in stroke-related activities, such as the number of strokes and average racket velocity. This indicates that the initial variations were observed in parameters linked to technical aspects of stroke execution. Our results also show that the tennis shot activities, such as the spin of the shots, are also strongly influenced by strategy, as balls were hit with higher spin on both the forehand and backhand sides in the defensive condition (*p* = 0.031; r = 0.90). The defensive strategy as a whole is characterized by tactics such as playing mostly from the baseline to the middle of the court without coming up to the net and avoiding hitting sharp angles in order to minimize wide shots [27]. In contrast, when using the offensive strategy, we can see from our results that there was significantly less running activity in terms of quantity (PlayerLoad, IMA CoD low right), which was also realized because, as we have seen from previous research, this strategy has a higher work/rest ratio, so there is less effective time to perform micromovements [12,27]. The tactical characteristics of the offensive strategy are completely opposite to the defensive strategy. First of all, offensive-style players like to run up to the net more and finish points there, and they use high-angle shots and vary them frequently. According to experts [12,27,54], attacking players also use higher velocity groundstrokes to finish points faster, but this is not supported by our results, as there was no significant difference in average speed for any of the groundstrokes between the test conditions we assessed (all *p* > 0.05). Our results, which contradict previous research regarding the velocity of groundstrokes, can be attributed to the fact that those studies examined non-elite and female tennis players [12,27]. At that level, the average shot velocity may have a greater influence on strategies and match outcomes compared to male players [13].

### 4.1. Practical Applications

From a practical point of view, the results of the present pilot study and the results of previous studies on similar topics suggest that the characteristics of each playing style and playing strategy should be taken into account from both a physical and a technical–tactical point of view when planning training sessions. The four most common playing styles (aggressive baseliner, counterpuncher, serve and volleyer, and all-court player) [54] are associated with a dominant offensive or defensive strategy conception [12]—the use of which naturally can depend on the opponent’s style, playing surface, weather conditions, and skill level [12,13,27]. In fact, for those types of players who are counterpunchers and all-court players, and prefer a predominantly defensive strategy in their matches, intermittent endurance capacities are the most important factor in conditioning training [1,12] because this strategy requires higher aerobic and anaerobic–lactic capacities due to longer ball movements and more acceleration in all three planes of motion (PlayerLoad). In contrast, the aggressive baseliner, serve-and-volleyer, and all-court player who prefers the offensive strategy is dominated by anaerobic–alactic energy production and explosive power capacity. According to Hettler and colleagues [55], when assessing conditional abilities, particularly endurance, and evaluating the results, the players’ preferred playing style should also be taken into account. It is worth categorizing players into separate endurance-based high-intensity interval training (HIIT) and sprint-based HIIT groups based on the so-called Speed Reserve Ratio (SRR) values, which can be calculated using the following formula: SRR = Maximal Sprint Speed (MSS)/Maximal Aerobic Speed (MAS) [56].

In technical–tactical training, the defensive strategy should focus on higher spin groundstrokes, baseline play, and minimizing unforced errors, while the offensive strategy should focus on higher angle and speed shots and more net play [54]. Naturally, in our research, we analyzed each playing strategy in controlled conditions rather than in open competition settings. However, since the players participating in this study were all-court players—meaning they effectively use both types of strategies in a mixed manner during matches—the data obtained here is relevant for analyzing official matches for both offensive and defensive strategies regardless of the instructions they received here. It should be noted, however, that since success in tennis is complex in all aspects (physical and technical–tactical), it is necessary to develop all the other skills in addition to those listed above for each strategy, especially for junior athletes, for whom long-term and complete development is the goal based on the LTAD (Long-Term Athlete Development) model [57]. Also, the periodization of training should take into account the different strategies of the players. For example, in the preparation period, in the periods between competitions or after an injury, the focus should be more on complex training, but in the competitive period, in order to provide more specific training, it is advisable to plan tennis players’ training sessions both on and off the court in a strategy and playing style specific way. In addition, in today’s modern tennis, monitoring of match data such as external and internal training load variables is now essential for the development of player profiles. Among other things, to be able to give players a more progressive load, which can reduce the risk of sports injuries. In fact, if we integrate these running activities, shot activities, and internal load variables we have used into a system like ACWR, this can be achieved under objective conditions, not only in the case of elite athletes but also in the case of recreational athletes [58].

### 4.2. Limitations and Advantages of This Study

A primary limitation of our research is the relatively small number of participants included in this study, and this may affect the generalizability of the results to the population of youth male tennis players, but it provides an important reference point as we only examined Hungarian elite tennis players, thus allowing us to investigate the load variables during high-quality simulated matches. Besides this, in our next research, we would like to use a within-subject design with repeated measures because this would have been ideal to assess individual variability more effectively. Nevertheless, the experimental protocol of the present pilot study will be extended to a larger sample in the future. In addition, we only focused on youth players in this study, but it is definitely worthwhile to look at elite adult tennis players in the future, as their data can be used to help coach junior players in the future. For running activities, it is also worth investigating variables that focus on both high-intensity accelerations and decelerations and monitoring of internal workload not only on a subjective RPE basis but also by measuring objective physiological responses. Furthermore, in the future, it would be worthwhile to examine shot spin using other smart tennis sensors, as the Zepp sensor we used showed only moderate agreement in detecting this type of shot.

However, the strength of our study is that we studied elite, ITF junior-ranked tennis players, two of whom had already had ATP ranking. Last but not least, this type of measurement has, to the best of our knowledge, only been used for women’s tennis players and appears here for the first time in men’s tennis.

## 5. Conclusions

In conclusion, we can say that the use of different playing strategies influences the spread of external load factors such as accelerations in three planes of motion and low-intensity change of directions in different directions, and also the spin of the groundstrokes. All of this allows for even more specific coaching for tennis players in terms of strategy and style of play both on and off the court. Specifically, counterpuncher-style players should focus more on developing aerobic and anaerobic–lactacid energy systems, while an aggressive baseliner, who primarily employs an offensive strategy in matches, should concentrate on anaerobic–alactacid processes and muscle power development during the majority of their training sessions. Last but not least, when analyzing matches, monitoring can be conducted by following the integrated methodology, always interpreting external and internal training load data in the context of the tactical situation rather than in isolation.

## Figures and Tables

**Figure 1 sports-13-00101-f001:**
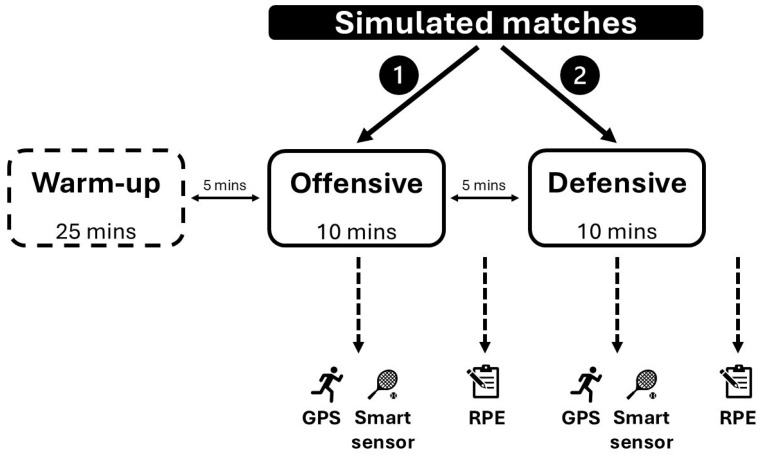
The schematic illustration of the experimental design. Legend: GPS = global positioning system; RPE = rating of perceived exertion.

**Figure 2 sports-13-00101-f002:**
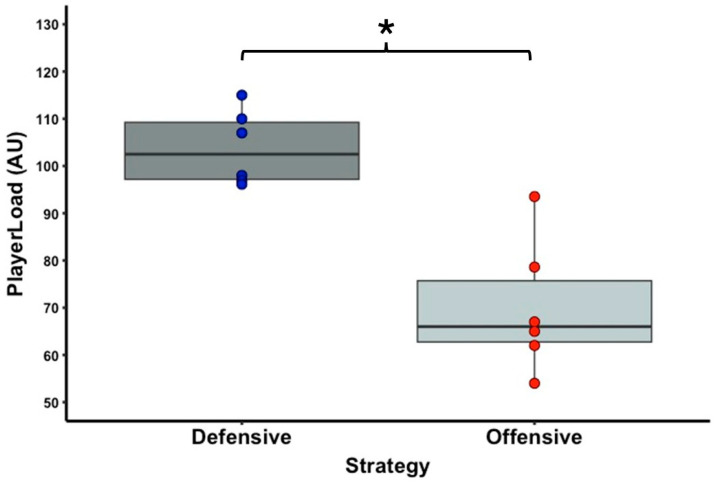
The significantly different PlayerLoad variable of the offensive and defensive strategy conditions. The boxplots show the median, the upper and lower quartiles, and the min and max values of the strategy conditions with individual data points. Legend: * indicates a significant difference between strategy conditions (*p* < 0.05).

**Figure 3 sports-13-00101-f003:**
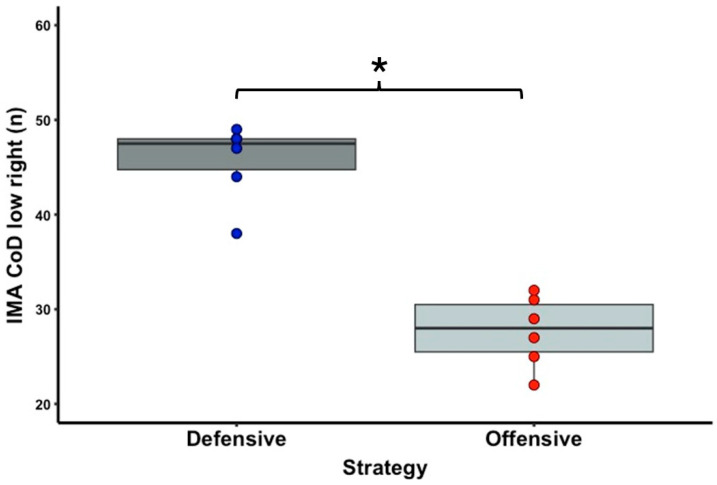
The significantly different IMA CoD low right variable of the offensive and defensive strategy conditions. The boxplots show the median, the upper and lower quartiles, and the min and max values of the strategy conditions with individual data points. Legend: * indicates a significant difference between strategy conditions (*p* < 0.05).

**Figure 4 sports-13-00101-f004:**
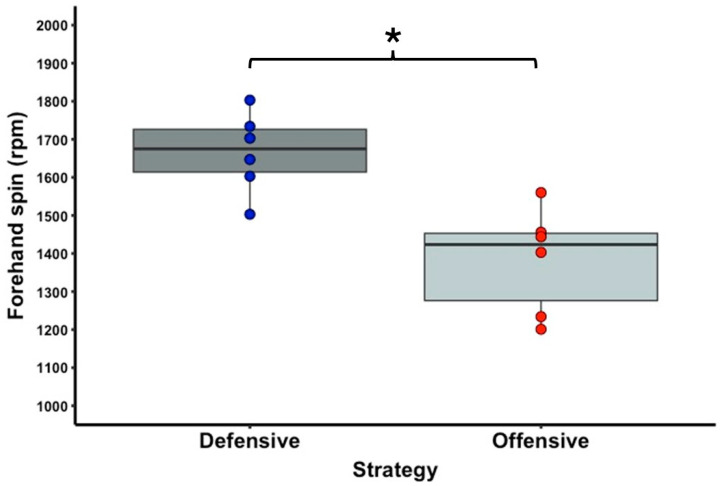
The significantly different forehand spin variable of the offensive and defensive strategy conditions. The boxplots show the median, the upper and lower quartiles, and the min and max values of the strategy conditions with individual data points. Legend: * indicates a significant difference between strategy conditions (*p* < 0.05).

**Figure 5 sports-13-00101-f005:**
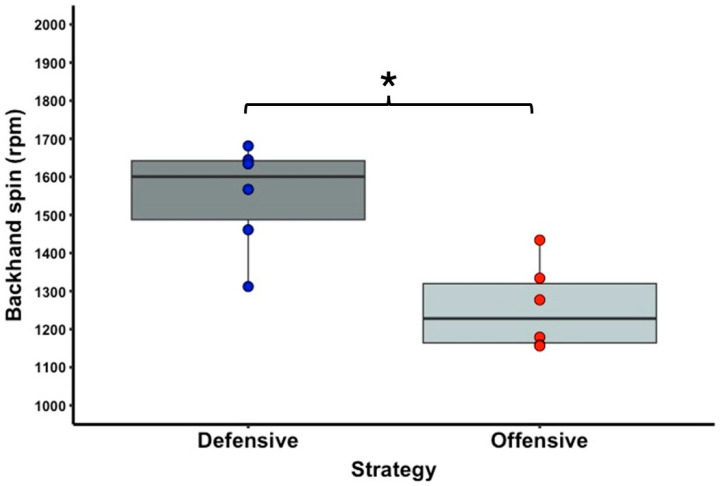
The significantly different backhand spin variable of the offensive and defensive strategy conditions. The boxplots show the median, the upper and lower quartiles, and the min and max values of the strategy conditions with individual data points. Legend: * indicates a significant difference between strategy conditions (*p* < 0.05).

**Table 1 sports-13-00101-t001:** External and internal load variable data of the offensive and defensive strategy conditions.

Variables	Offensive		Defensive					
	Mean ± SD	95% CI	Mean ± SD	95% CI	*T*	*Z*	*p*	*r*
**Running activities**								
PlayerLoad (AU)	70.0 ± 14.0	55.32–84.73	103.8 ± 7.9	95.51–112.16	0.0	2.201	0.031 *	0.90
IMA CoD low right (n)	27.7 ± 3.8	23.70–31.63	45.7 ± 4.1	41.33–50.00	0.0	2.201	0.031 *	0.90
IMA CoD low left (n)	43.7 ± 8.5	34.77–52.56	58.9 ± 9.0	49.41–68.25	2.0	1.782	0.090	0.73
IMA CoD high right (n)	1.3 ± 0.8	0.48–2.19	4.1 ± 4.9	0.95–9.28	3.0	1.572	0.140	0.64
IMA CoD high left (n)	6.7 ± 2.0	4.60–8.73	5.3 ± 1.0	4.25–6.42	12.0	1.214	0.279	0.50
**Shot activities**								
Forehand velocity (km·h^−1^)	100.2 ± 2.2	97.83–102.51	99.8 ± 3.0	96.69–102.98	6.0	0.365	0.855	0.15
Backhand velocity (km·h^−1^)	98.0 ± 4.9	92.86–103.14	93.5 ± 3.9	90.27–96.73	16.0	1.153	0.313	0.47
Forehand spin (rpm)	1383.0 ± 138.7	1237.51–1528.49	1665.5 ± 105.5	1554.78–1776.22	0.0	2.201	0.031 *	0.90
Backhand spin (rpm)	1256.5 ± 112.7	1138.19–1374.81	1550.0 ± 140.1	1403.01–1696.99	0.0	2.201	0.031 *	0.90
**Internal load**								
RPE (AU)	5.3 ± 0.8	4.48–6.19	6.7 ± 1.4	5.23–8.10	1.5	1.633	0.102	0.67

Legend: CI = confidence interval; IMA = inertial movement analysis; CoD = change of direction; RPE = rating of perceived exertion; * indicates a significant difference between strategy conditions (*p* < 0.05).

## Data Availability

The data presented in this study are available on request from the corresponding author.

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
