# Peer review of "Comparison of External and Internal Training Loads in Elite Junior Male Tennis Players During Offensive vs. Defensive Strategy Conditions: A Pilot Study"

_sports, 2025, doi:10.3390/sports13040101_

Round 1

Reviewer 1 Report

Comments and Suggestions for Authors

Dear corresponding Author, thank you for submitting your work to Sports journal and congratulations on your research.

Brief Summary The study examined the effects of offensive and defensive strategies on external and internal load parameters in 6 elite junior male tennis players. Players performed 10-minute simulated matches with both strategies. External load parameters were measured.

General Comments The work addresses a current topic in tennis performance, with an approach that considers both technical-tactical and physical aspects. At first reading it seemed a well-conceived work, however, there are some methodological limitations that deserve attention:

  • The extremely small sample (n=6) constitutes in my opinion the main limitation of the study. Although the authors define it as a pilot study, it would be appropiate to discuss more deeply how this limitation influences the interpretation of results and their generalisability.
  • The duration of simulated matches (10 minutes) might not adequately reflect real game conditions. It would be useful to justify this methodological choice, perhaps comparing it with the average duration of sets in youth tennis and supporting it with greater scientific consistency.
  • There is a lack of in-depth discussion on how the results compare with game characteristics observed in real competition conditions, especially considering that players were aware of strategic instructions. I don't see particular transferability with this approach, but I would like clarification.

Specific Comments

  • Lines 49-50: When you state "an average of 4 to 6 change of directions per point", I didn't understand whether this data refers to total changes of direction or per player.
  • Lines 89-91: The cross-sectional experimental design might not be ideal for evaluating individual differences in responses to different strategies. Why didn't you consider a within-subject design with repeated measures? I would like clarification.
  • Lines 134-137: The instructions given for the two strategies ("try to win the point by yourself" vs "try to win the point by reducing unforced errors") seem rather generic to me. Players could have interpreted them differently. Did you verify understanding of the instructions and adherence to them during matches? Also, were they given in English or Spanish?
  • Table 1: The data show rather high standard deviation values in some parameters (e.g., IMA CoD high right in defensive condition: 4.1 ± 4.9), suggesting considerable variability among subjects that would deserve more in-depth discussion.
  • Lines 308-312: When you state that offensive players use higher velocity shots, but your data don't support this, it would be useful to explore this discrepant aspect compared to existing literature instead of simply hypothesizing that it's related to age, gender or skill level.
  • Line 336: Given the numerous criticalities, I belive a dedicated "Limitations" section is necesary where all points are clarified.

Overall, the study offers interesting insights into understanding the impact of different game strategies on load parameters in elite youth tennis, but needs some revisions to improve its methodological rigor and practical applicability. I believe it's necessary for me to wait to read an improved version to give a final decision.

Author Response

Dear Reviewer,

Thank you very much for your work and valuable comments. Below are our responses to each comment, summarized in bullet points.

Comments 1: 

  • The extremely small sample (n=6) constitutes in my opinion the main limitation of the study. Although the authors define it as a pilot study, it would be appropiate to discuss more deeply how this limitation influences the interpretation of results and their generalisability.

Response 1:

Thank you very much for pointing this out. We totally agree with your comment. The generalisation of our results should definitely be approached with caution, as the sample size is small. Therefore, it cannot be generalised to the entire youth male tennis player population in this pilot study, but it provides a solid foundation for future research. We have revised it in the manuscript (pp. 12., lines: 445-448 ), and we have highlighted it in red colour. 

Comments 2: 

  • The duration of simulated matches (10 minutes) might not adequately reflect real game conditions. It would be useful to justify this methodological choice, perhaps comparing it with the average duration of sets in youth tennis and supporting it with greater scientific consistency.

Response 2:

Thank you very much for pointing this out. We totally agree with your comment. The 10-minute intervals were chosen because they allowed the players to play multiple matches in one day without the influence of accumulating fatigue, furthermore the players were between two tournaments and could not be overloaded, as they are professional athletes and they were in-season. We have revised it in the manuscript (pp. 4., lines: 162-165 ), and we have highlighted it in red colour. 

Comments 3: 

  • There is a lack of in-depth discussion on how the results compare with game characteristics observed in real competition conditions, especially considering that players were aware of strategic instructions. I don't see particular transferability with this approach, but I would like clarification.

Response 3:

We absolutely agree with your comment in this topic. We analyzed each playing strategy in controlled conditions rather than in open competition settings. However, since the players participating in the study were all-court players—meaning they effectively use both types of strategies in a mixed manner during matches—the data obtained here is relevant for analyzing official matches for both offensive and defensive strategies regardless of the instructions they received here. But we revised it in the manuscript (pp. 12., lines: 426-431 ), and we have highlighted it in red colour. 

Comments 4:

  • Lines 49-50: When you state "an average of 4 to 6 change of directions per point", I didn't understand whether this data refers to total changes of direction or per player.

Response 4:

Thank you very much, your insight is absolutely valid. These data refers per player, but we revised it in the manuscript (pp. 2., lines: 54-55) and we have highlighted it in red colour. 

Comments 5:

  • Lines 89-91: The cross-sectional experimental design might not be ideal for evaluating individual differences in responses to different strategies. Why didn't you consider a within-subject design with repeated measures? I would like clarification.

Response 5:

Thank you very much for pointing this out. The cross-sectional design was chosen due to the limited number of participants available for the study in this elite population and the time constraints involved. A within-subject design with repeated measures would have been ideal to assess individual variability more effectively - we add this sentence to the 'Limitations' sub-section in the manuscript (pp. 12., lines: 448-450 ) and we have highlighted it in red colour.  However, we intend to conduct further research with a more controlled approach to investigate this aspect more comprehensively.

Comments 6:

  • Lines 134-137: The instructions given for the two strategies ("try to win the point by yourself" vs "try to win the point by reducing unforced errors") seem rather generic to me. Players could have interpreted them differently. Did you verify understanding of the instructions and adherence to them during matches? Also, were they given in English or Spanish?

Response 6: 

Thank you very much for your comment. In the main text, we mentioned that at halftime, the coaches reminded the players to follow the given instructions. We have now highlighted this in red (pp. 4., lines: 184-187). Additionally, the instructions were clear to the players, as they had encountered them multiple times during their regular tactical training sessions. Furthermore, the instructions were given in Hungarian, as the players were of Hungarian nationality - we have also revised it in the manuscript (pp. 4., lines: 175-179) and we have highlighted it in red colour.

Comments 7:

  • Table 1: The data show rather high standard deviation values in some parameters (e.g., IMA CoD high right in defensive condition: 4.1 ± 4.9), suggesting considerable variability among subjects that would deserve more in-depth discussion.

Response 7:

Thank you very much, your insight is absolutely valid. when examining the IMA CoD right high variable, relatively higher standard deviation values were found despite the small sample size, which we believe may have arisen due to the participants having different anthropometric characteristics (body height and body mass), which can significantly influence the development of high-intensity actions. We have revised it in the manuscript (pp. 10-11., lines: 357-361 ) and we have highlighted it in red colour.

Comments 8: 

  • Lines 308-312: When you state that offensive players use higher velocity shots, but your data don't support this, it would be useful to explore this discrepant aspect compared to existing literature instead of simply hypothesizing that it's related to age, gender or skill level.

Response 8: 

Thank you very much for your comment. We add some revision to this topic with references (pp. 11., lines: 395-402) and we have highlighted it in red colour. Tha main point is that previous research regarding the velocity of groundstrokes, can be attributed to the fact that those studies examined non-elite and female tennis players, and the average shot velocity may have a greater influence on strategies and match outcomes compared to male players.

Comments 9:

  • Line 336: Given the numerous criticalities, I belive a dedicated "Limitations" section is necesary where all points are clarified.

Response 9:

Thank you very much for pointing this out. We totally agree with your comment, and as we mentioned it in the 'Response 1', we add a 'Limitations and advantages of the study' sub-section (pp. 12., lines: 443) and we have highlighted in red. 

Finally, I would like to thank you very much for the many useful criticisms and advice, which have definitely helped me to produce a much more demanding and scientifically sound work.

Kind regards,

Péter János Tóth and the co-authors

Reviewer 2 Report

Comments and Suggestions for Authors

Dear Editor,

Thank you for inviting me to review the manuscript entitled "Comparison of external and internal training loads in elite junior male tennis players during offensive vs. defensive strategy conditions: a pilot study" (ID: sports-3520486). This study examined the effects of offensive and defensive strategy conditions on external training load variables (running activities measured with GPS sensors and tennis shot activities with smart sensors) and internal training load variables (measured subjectively using the RPE method) in elite junior male tennis players. The main findings revealed that players using a defensive strategy demonstrated significantly higher values for PlayerLoad™, IMA CoD low right, forehand spin, and backhand spin compared to when using an offensive strategy, while no significant differences were observed in other external and internal load parameters.

Overall, this manuscript makes a valuable contribution to the sports science literature, particularly in the field of tennis performance analysis, by examining the impact of different playing strategies on physiological and technical parameters. The methodology is generally sound, and the findings provide useful insights for coaching and training practices. However, there are several areas that could be strengthened to improve the clarity, rigor, and practical implications of the manuscript.

General Comments

  1. Introduction: The literature review provides adequate background on tennis match characteristics and the integrated approach to match analysis. However, the framework connecting strategy conditions to training loads could be better established, particularly regarding the theoretical basis for the study's hypotheses.
  2. Methods: The experimental approach is clearly described, but some methodological details require clarification, particularly regarding the randomization process for strategy assignment and the validity of the measurement tools used.
  3. Results: The results are generally well-presented, though the figures could be enhanced for better interpretation. Additionally, a more comprehensive analysis of the practical significance of the findings would strengthen this section.
  4. Discussion: The discussion addresses the main findings but could be expanded to more critically evaluate the practical implications of the results, particularly regarding training prescription for different playing strategies.
  5. Limitations: The authors acknowledge the small sample size as a limitation, but a more comprehensive discussion of other methodological limitations would strengthen the scientific rigor of the manuscript.

Specific Comments

Introduction

  • The connection between playing strategies (offensive vs. defensive) and their impact on physiological demands is mentioned but could be strengthened with more specific literature on how tactical decisions influence external and internal loads in tennis or other racket sports.
  • The authors state two hypotheses at the end of the introduction: (i) a difference in external load variables between strategies, and (ii) no significant difference in internal load factors. The theoretical basis for the second hypothesis is not clearly explained and seems contradictory to some of the literature cited earlier about differences in physiological demands based on strategies.
  • The definition of offensive and defensive strategies in tennis could be more clearly established in the introduction to provide context for readers who may not be familiar with tennis-specific terminology.

Methods

  • The sample size (n=6) is small, which is acknowledged as this is a pilot study. However, a power analysis or justification for this sample size would strengthen the methodology.
  • The description of the players' characteristics is thorough, but information on their typical playing styles (whether they are naturally offensive or defensive players) would be valuable context for interpreting the results.
  • The simulated match protocol (10-minute blocks) is relatively short compared to actual tennis matches. A rationale for this timeframe and a discussion of its potential impact on the results would be beneficial.
  • The strategy instructions given to players were relatively simple. More detailed information on how these instructions were reinforced during play and whether compliance was monitored would strengthen the methodology.
  • The validity and reliability of the Zepp Tennis smart sensor for measuring spin parameters should be more thoroughly addressed, as this is critical for interpreting the findings related to forehand and backhand spin.

Results

  • Table 1 presents comprehensive data, but visually separating the different categories of variables (running activities, tennis shot activities, and internal load) would enhance readability.
  • In Figures 2 and 3, adding individual data points alongside the boxplots would provide more information about the distribution and variability within this small sample.
  • The statistical approach is appropriate for the non-parametric data, but the authors should consider including confidence intervals or additional effect size interpretations to better contextualize the magnitude of the differences observed.
  • The non-significant trend toward higher RPE in defensive strategy (p = 0.102; r = 0.67) shows a large effect size but was not discussed in detail. This merits more attention given the small sample size and potential practical significance.

Discussion

  • The discussion appropriately contextualizes the findings within the existing literature but could more critically evaluate why some hypothesized differences were observed while others were not.
  • The authors note that "it is always the technique-tactics that will be affected first by the application of different strategies, and only then the tennis shot and running activities, and then, at the very end, the internal load" (lines 293-297). This conceptual framework is interesting but would benefit from more empirical support.
  • The practical applications section provides valuable insights for training prescription based on playing strategy, but could be expanded with more specific recommendations for developing training programs that account for the different physiological demands of offensive and defensive play.
  • The discussion of why some variables (e.g., PlayerLoad™ and spin parameters) showed significant differences while others (e.g., velocities and RPE) did not could be more thoroughly explored.

Conclusion

  • The conclusion appropriately summarizes the main findings but could be strengthened by more explicitly addressing the implications for future research and practice.
  • The concluding paragraph mentions the potential for "even more specific coaching for tennis players in terms of strategy and style of play both on and off the court" but specific recommendations are limited.

In conclusion, this manuscript presents novel and valuable findings regarding the effects of different strategy conditions on external and internal training loads in elite junior male tennis players. With the suggested revisions, particularly regarding the theoretical framework, methodological clarity, and practical implications, this manuscript will make a significant contribution to the scientific literature on tennis performance analysis and training prescription.

Best regards,

The Reviewer

Author Response

Dear Reviewer,

Thank you very much for your work and valuable comments. Below are our responses to each comment, summarized in bullet points.

Comments 1:

  • The connection between playing strategies (offensive vs. defensive) and their impact on physiological demands is mentioned but could be strengthened with more specific literature on how tactical decisions influence external and internal loads in tennis or other racket sports.

Response 1:

Thank you very much for your comments, and for pointing this. We have strengthened it with one more literature in racket sports (table tennis). We revised it in the manuscript (pp. 3., lines: 97-101) and we have highlighted it in red. 

Comments 2:

  • The authors state two hypotheses at the end of the introduction: (i) a difference in external load variables between strategies, and (ii) no significant difference in internal load factors. The theoretical basis for the second hypothesis is not clearly explained and seems contradictory to some of the literature cited earlier about differences in physiological demands based on strategies.

Response 2:

Thank you, it is absolutely valid statement. We believe – thanks to your previous comment – that with the additional publication we mentioned in the previous section, the results can no longer be fully contradicted, as it was also studied in a racket sport. According to our current knowledge, there are no other similar studies in regular tennis, which is why we included one from another racket sport. Furthermore, we have clarified our second hypothesis by adding the subjective RPE method to make it clearer what we mean by the internal load factor. We revised it in the manuscript (pp. 3., lines: 110-112) and we have highlighted it in red. 

Comments 3:

  • The definition of offensive and defensive strategies in tennis could be more clearly established in the introduction to provide context for readers who may not be familiar with tennis-specific terminology.

Response 3:

Thank you very much for your comments, and for pointing this. It is absolutely valid. In the main text, we have added a paragraph where we briefly describe the two game strategies (pp. 2., lines: 64-70). 

Comments 4:

  • The sample size (n=6) is small, which is acknowledged as this is a pilot study. However, a power analysis or justification for this sample size would strengthen the methodology.

Response 4:

Thank you very much. We have added som sentence in the manuscript (pp. 3., lines: 129-132). 

Comments 5:

  • The description of the players' characteristics is thorough, but information on their typical playing styles (whether they are naturally offensive or defensive players) would be valuable context for interpreting the results.

Response 5:

Thank you so much your comments. We mentioned it in the inclsuion criteria that (iv) ability to apply both offensive and defensive play (lines: 126-127). However, for the sake of clarity, we have added an extra sentence to this effect (pp. 3., lines: 139-141). 

Comments 6:

  • The simulated match protocol (10-minute blocks) is relatively short compared to actual tennis matches. A rationale for this timeframe and a discussion of its potential impact on the results would be beneficial.

Response 6:

Thank you very much for pointing this. The 10-minute intervals were chosen because they allowed the players to play multiple matches in one day without the influence of accumulating fatigue, furthermore the players were between two tournaments and could not be overloaded, as they are professional athletes and they were in-season. We have added this paragraph to the manuscript (pp. 4., lines: 160-165).

Comments 7:

  • The strategy instructions given to players were relatively simple. More detailed information on how these instructions were reinforced during play and whether compliance was monitored would strengthen the methodology.

Response 7:

Thank you so much this information. Now we have defined it more clearly in the manuscript (pp. 4., lines: 175-187). 

Comments 8:

  • The validity and reliability of the Zepp Tennis smart sensor for measuring spin parameters should be more thoroughly addressed, as this is critical for interpreting the findings related to forehand and backhand spin.

Response 8:

Thank you very much for your comments. You are absolutely right. The shot velocity measurements from the Zepp sensor showed an almost perfect agreement with the gold standard VICON system (ICC = 0.983; p < 0.001), demonstrating excellent validity for speed measurement. In addition, this type of smart sensor showed moderate agreement with actual shot types (κ = 0.612), like the spin parameters. We have added this statement to the text (pp. 6., lines: 227-231), and we have highlighted in red. Furthermore, we have also added some statements regarding this to the 'Limitations' sub-section, but we have also highlighted in red (pp. 12-13, lines: 457-459).

Comments 9:

  • Table 1 presents comprehensive data, but visually separating the different categories of variables (running activities, tennis shot activities, and internal load) would enhance readability.

Response 9:

Thank you so much, it is a really good idea. We revised the Table 1 based on your opinions (pp.7, Table 1.). 

Comments 10:

  • In Figures 2 and 3, adding individual data points alongside the boxplots would provide more information about the distribution and variability within this small sample.

Response 10:

Thank you very much for your valuable guidance. On the one hand, we took your advice and added individual data points to each boxplot. On the other hand, we separated the figures for better visualization (pp. 8-9., Figure 2-5).

Comments 11:

  • The statistical approach is appropriate for the non-parametric data, but the authors should consider including confidence intervals or additional effect size interpretations to better contextualize the magnitude of the differences observed.

Response 11:

Thank you for your opinions. We have added the confidence intervals (95% CI) to the Table 1 (pp. 7, Table 1.).

Comments 12:

  • The non-significant trend toward higher RPE in defensive strategy (p = 0.102; r = 0.67) shows a large effect size but was not discussed in detail. This merits more attention given the small sample size and potential practical significance.

Response 12:

Absolutely agree with this statement. This could mean that, despite the lack of a statistically significant difference between the two game strategies in the RPE variable, it is clear that players became more fatigued in the defensive strategy condition. This is supported by the study of Bernardi and colleagues [13], which found that defensive players have the highest effective playing time, with a 38.5% higher value, which may be correlated with the development of internal load. We have added this information to the manuscript in red colour (pp. 11., lines: 361-369).

Comments 13:

  • The discussion appropriately contextualizes the findings within the existing literature but could more critically evaluate why some hypothesized differences were observed while others were not.

Response 13:

Thank you so much. The significant changes are explained as follows in the document:

"Our results showed that for the defensive strategy condition, significantly higher values were obtained for the PlayerLoad variable, which implies that players in this strategy perform more accelerations in all three planes of motion than in the offensive strategy, and higher values for the IMA CoD low right parameter. This suggests that, since all participants were right-handed in the research, and considering that in modern tennis, more forehand strokes are made than backhands, a statistically significant difference in favor of the defensive strategy occurred in this IMA CoD low right variable. This is because, in this parameter, rallies tend to be longer, and therefore, more shots are played." (pp.10., lines: 343-352). 

Furthermore, the non-significant differences are explained as follows:

"Although there was no statistically significant effect of the different strategies on the other external load variables, large or medium effect sizes were still observed. The running activities in which there was no statistically significant difference can be explained by the fact that, for example, in the two high-intensity direction changes, the players already achieved low values, which are negligible compared to the low-intensity movements." (pp.10., lines: 352-357).

"According to experts attacking players also use higher velocity groundstrokes to finish points faster, but this is not supported by our results, as there was no significant difference in average speed for any of the groundstrokes between the test conditions we assessed (all p > 0.05). Our results, which contradict previous research regarding the velocity of groundstrokes, can be attributed to the fact that those studies examined non-elite and female tennis players. At that level, the average shot velocity may have a greater influence on strategies and match outcomes compared to male players." (pp.11., lines: 395-402).

Comments 14:

  • The authors note that "it is always the technique-tactics that will be affected first by the application of different strategies, and only then the tennis shot and running activities, and then, at the very end, the internal load" (lines 293-297). This conceptual framework is interesting but would benefit from more empirical support.

Response 14:

Thank you very much for the valuable insight once again. We have supported this conceptual framework in the text with the following research findings: "This statement is supported by the research findings of Hoppe and colleagues, who found that the passive, active, and mixed playing strategy conditions induce large effects on external loads (running distances with high acceleration and deceleration), moderate effects on internal loads (energy expenditures with high metabolic power, lactate concentration, and rating of effort), and very large effects on technical-tactical actions (number of ground strokes and errors) and activity profiles (strokes per rally, rally duration, work to rest ratio, and effective playing time). (pp. 11., lines: 372-379).

Comments 15:

  • The practical applications section provides valuable insights for training prescription based on playing strategy, but could be expanded with more specific recommendations for developing training programs that account for the different physiological demands of offensive and defensive play.

Response 15:

Thank you very much for the valuable comment and insight. In the new 'Practical applications' sub-section in red (pp.11., lines: 403-423) we have already expanded on this knowledge, supporting it with two additional references.

Comments 16-17:

  • The conclusion appropriately summarizes the main findings but could be strengthened by more explicitly addressing the implications for future research and practice.
  • The concluding paragraph mentions the potential for "even more specific coaching for tennis players in terms of strategy and style of play both on and off the court" but specific recommendations are limited.

Response 16-17:

Finally, and most importantly, we sincerely appreciate the suggestions and advice provided for the conclusion section. We fully agree with them. To address this, we have supplemented this section in the text, highlighting the additions in red as follows:

" For example counterpuncher style players should focus more on developing aerobic and anaerobic-lactacid energy systems, while an aggressive baseliner, who primarily employs an offensive strategy in matches, should concentrate on anaerobic-alactacid processes and muscle power development during the majority of their training sessions. Last but not least, when analyzing matches, monitoring can be conducted by strategy following the integrated methodology, always interpreting external and internal training load data in the context of the tactical situation rather than in isolation." (pp.13., lines: 469-476).

Additionally, within the discussion section, we have created separate sub-sections titled "Practical Applications" (pp.11., lines: 403) and "Limitations and Advantages of the Study" (pp.12., lines: 443) This was done to provide training methodology recommendations based on our findings and to better highlight the study’s limitations while drawing attention to potential future research directions. 

Finally, I would like to thank you very much for the many useful criticisms and advice, which have definitely helped me to produce a much more demanding and scientifically sound work.

Kind regards,

Péter János Tóth and the co-authors

Reviewer 3 Report

Comments and Suggestions for Authors

First of all, I would like to thank the editors of the journal for the opportunity to collaborate in the review of this manuscript. The article shows an intervention to analyse the level of external and internal loading in junior tennis players after playing two simulated matches.

First of all, I would like to comment that one of the major limitations of this article is the small sample size of the study. In total, this research work is carried out with only six tennis players, who only play a total of two simulated matches. Between the size of the sample and the small number of matches analysed, it is understandable that the authors are unable to draw rigorous conclusions about the object of study. 

For this reason, journal editors are advised to consider whether the article is sufficiently scientifically rigorous to be considered publishable.

However, in order to improve the research carried out, a number of improvements are recommended for this scientific work:

  1. In the keywords, words used in the title of the article should not be added.
  2. In the abstract the instruments used to collect the data should be specified.
  3. In the introduction, it is recommended to define the main objective of the research, as well as to add the hypothesis under study. In addition, a further review of the scientific literature is recommended, as the authors describe that there are no studies that have compared different strategies to measure external and internal load in tennis players, but different types of training have been used to do so, which have not been cited in this study, such as, for example: Rodríguez-Cayetano, A., Martín, Ó., Hernández-Merchán, F., & Pérez-Muñoz, S. (2022). Internal and external load in competitive tennis: comparison of three types of training. Retos44, 534–541. https://doi.org/10.47197/retos.v44i0.90583
  4. In the methodology, it should be better specified what is meant by an offensive and defensive strategy, as it would be advisable to establish patterns of play that the players have taken into account, not just not missing the ball or winning the point.
  5. Once the scientific literature has been reviewed, a revision of the discussion is recommended, adding research such as that cited above, improving the discussion of the results obtained.

Author Response

Dear Reviewer,

Thank you very much for your work and valuable comments. Below are our responses to each comment, summarized in bullet points.

Comments 1:

  • First of all, I would like to comment that one of the major limitations of this article is the small sample size of the study. In total, this research work is carried out with only six tennis players, who only play a total of two simulated matches. Between the size of the sample and the small number of matches analysed, it is understandable that the authors are unable to draw rigorous conclusions about the object of study. 

Response 1:

Thank you very much for your valuable insight. We fully agree with your points; however, in this case, we were unable to include more participants in the study due to the inclusion and exclusion criteria. Hungary is a relatively small country and not a major tennis nation, which limited our pool of eligible players. Nevertheless, our research is strengthened by the fact that, although only six players participated, they represent the best of the Hungarian youth tennis scene, with two of them already holding ATP world rankings, confirming their status as professional players. Based on your observations, we have supplemented the manuscript with the following statements, which we have highlighted in red:

"We have to mention that after adhering to the exclusion criteria there are relatively small number of participants included in the study, but our study is strengthened by the fact that the final participants were elite junior or already professional-level tennis players, ensuring that very high-quality matches took place." (pp.3., lines: 129-132, section: 2.2 Participants).

"A primary limitation of our research is the relatively small number of participants included in the study, and this may affect the generalisability of the results to the population of youth male tennis players, but it provides an important reference point as we only examined Hungarian elite tennis players, thus allowing us to investigate the load variables during high-quality simulated matches." (pp.12., lines: 444-448, 4.2 Limitations and advantages of the study).

Comments 2:

  1. In the keywords, words used in the title of the article should not be added.

Response 2:

Thank you very much for your insight. We have taken your advice into account and replaced the keywords "tennis" and "strategy" with "racket sports" and "playing style" (pp. 1., Keywords).

Comments 3:

2. In the abstract the instruments used to collect the data should be specified.

Response: 3:

Thank you very much. We did not specify the exact types of devices used in the abstract because, firstly, the abstract has a limited word count (200 words), and we wanted to prioritize summarizing more essential information in this section. Secondly, based on our review of similar studies, it is common practice to mention only general details in the abstract. However, we have taken your advice into account and have now specified the frequency at which the GPS sensor operated, as we believe this is a relevant detail (pp. 1., Abstract).

Comments 3:

3. In the introduction, it is recommended to define the main objective of the research, as well as to add the hypothesis under study. In addition, a further review of the scientific literature is recommended, as the authors describe that there are no studies that have compared different strategies to measure external and internal load in tennis players, but different types of training have been used to do so, which have not been cited in this study, such as, for example: Rodríguez-Cayetano, A., Martín, Ó., Hernández-Merchán, F., & Pérez-Muñoz, S. (2022). Internal and external load in competitive tennis: comparison of three types of training. Retos44, 534–541. https://doi.org/10.47197/retos.v44i0.90583

Response 3:

Thank you very much for your valuable feedback. Based on your suggestion, we have added a clear statement of the main research objective and the hypotheses under study in the 'Introduction' section (pp.3., lines: 105-112). Furthermore, we have expanded the literature review by incorporating additional relevant studies (Milioni et al., 2018) (pp.3., lines:97-101). Specifically, we have now cited the study by Rodríguez-Cayetano et al. (2022), as it provides valuable insights into the comparison of different types of training in assessing internal and external load in competitive tennis players (pp. 2., lines: 89-93). Furthermore, the literature you mentioned is so relevant that we were able to cite it in another sentence within the 'Variables' subsection as well (pp. 6., lines: 232). These additions help to further contextualize our research within the existing body of knowledge.

Comments 4:

4. In the methodology, it should be better specified what is meant by an offensive and defensive strategy, as it would be advisable to establish patterns of play that the players have taken into account, not just not missing the ball or winning the point.

Response 4: 

Thank you very much for your valuable insight. Based on this, we have added a few lines to the 'Introduction' section, in which we describe the tactical situations and technical manifestations most characteristic of each strategy, according to previous publications (pp.2., lines: 64-70). Additionally, we also discuss these strategic instructions in the 'Procedures' subsection (pp.4., lines: 175-177): 

"With these brief yet concise instructions, the players applied the tactical situations characteristic of the respective strategies, as mentioned earlier [14]. These instructions were given in hungarian language, as the players were of Hungarian nationality, furthermore the instructions were clear to the players, as they had encountered them multiple times during their regular tactical training sessions."

Comments 5:

5. Once the scientific literature has been reviewed, a revision of the discussion is recommended, adding research such as that cited above, improving the discussion of the results obtained.

Response 5:

Thank you for the valuable feedback. We fully agree with your suggestion. After reviewing the relevant scientific literature, we have revised the discussion section to incorporate the study you mentioned (pp. 11., lines: 379-383). We believe that adding this reference helps to better contextualize our findings and enriches the analysis of the results. 

Finally, I would like to thank you very much for the many useful criticisms and advice, which have definitely helped me to produce a much more demanding and scientifically sound work.

Kind regards,

Péter János Tóth and the co-authors

Round 2

Reviewer 1 Report

Comments and Suggestions for Authors

I have carefully read the changes made by the authors in response to my comments. I believe that the work is still weak in some areas, but they have sufficiently clarified the remaining critical issues in the limitations. Therefore, I have no objections to recommending its publication in its current form.

Reviewer 2 Report

Comments and Suggestions for Authors

Dear Editor,

I have reviewed the revised manuscript entitled "Comparison of external and internal training loads in elite junior male tennis players during offensive vs. defensive strategy conditions: a pilot study" (ID: sports-3520486) along with the authors' responses to my initial review.

I am pleased to report that the authors have satisfactorily addressed all the concerns and suggestions raised in my previous review. The revisions have significantly improved the manuscript in several key areas:

  1. Theoretical Framework: The authors have strengthened the connection between playing strategies and physiological demands with additional literature from racket sports, providing better context for their hypotheses.
  2. Methodological Clarity: The authors have:
    • Added justification for the sample size
    • Clarified players' typical playing styles
    • Provided rationale for the 10-minute match protocol
    • Enhanced the description of strategy instructions and compliance monitoring
    • Addressed the validity and reliability of the Zepp Tennis smart sensor
  3. Results Presentation: The presentation of results has been improved by:
    • Visually separating different categories of variables in Table 1
    • Adding individual data points to the boxplots and separating figures for better visualization
    • Including confidence intervals to better contextualize the magnitude of differences
  4. Discussion and Implications: The discussion now:
    • More thoroughly addresses non-significant findings with large effect sizes
    • Provides empirical support for the conceptual framework
    • Includes expanded practical applications with specific training recommendations
    • Features a strengthened conclusion with clearer implications for practice

The manuscript now makes a valuable contribution to the scientific literature on tennis performance analysis, particularly regarding the effects of different strategy conditions on external and internal training loads in elite junior male tennis players. The findings provide useful insights for coaching and training practices, with clear practical applications.

Based on these improvements, I recommend that the manuscript be accepted for publication.

Sincerely,

The Reviewer

Reviewer 3 Report

Comments and Suggestions for Authors

The authors have made the suggested changes to improve the manuscript. However, there is still a major problem with the number of subjects, so, if the editors have no objection, the article can be published.